# The Role of Methionine-Rich Diet in Unhealthy Cerebrovascular and Brain Aging: Mechanisms and Implications for Cognitive Impairment

**DOI:** 10.3390/nu15214662

**Published:** 2023-11-03

**Authors:** Anna Ungvari, Rafal Gulej, Boglarka Csik, Peter Mukli, Sharon Negri, Stefano Tarantini, Andriy Yabluchanskiy, Zoltan Benyo, Anna Csiszar, Zoltan Ungvari

**Affiliations:** 1Department of Public Health, Semmelweis University, 1089 Budapest, Hungary; 2Vascular Cognitive Impairment, Neurodegeneration and Healthy Brain Aging Program, Department of Neurosurgery, University of Oklahoma Health Sciences Center, Oklahoma City, OK 73104, USA; rafal-gulej@ouhsc.edu (R.G.); boglarka-csik@ouhsc.edu (B.C.); peter-mukli@ouhsc.edu (P.M.); sharon-negri@ouhsc.edu (S.N.); stefano-tarantini@ouhsc.edu (S.T.); andriy-yabluchanskiy@ouhsc.edu (A.Y.); anna-csiszar@ouhsc.edu (A.C.); zoltan-ungvari@ouhsc.edu (Z.U.); 3Oklahoma Center for Geroscience and Healthy Brain Aging, University of Oklahoma Health Sciences Center, Oklahoma City, OK 73104, USA; 4International Training Program in Geroscience, Department of Public Health, Doctoral School of Basic and Translational Medicine, Semmelweis University, 1089 Budapest, Hungary; 5Stephenson Cancer Center, University of Oklahoma, Oklahoma City, OK 73104, USA; 6Department of Health Promotion Sciences, College of Public Health, University of Oklahoma Health Sciences Center, Oklahoma City, OK 73104, USA; 7Institute of Translational Medicine, Semmelweis University, 1094 Budapest, Hungary; benyo.zoltan@med.semmelweis-univ.hu; 8Cerebrovascular and Neurocognitive Disorders Research Group, Eötvös Loránd Research Network, Semmelweis University, 1094 Budapest, Hungary; 9International Training Program in Geroscience, Department of Translational Medicine, Doctoral School of Basic and Translational Medicine, Semmelweis University, 1089 Budapest, Hungary

**Keywords:** endothelial dysfunction, cerebral blood flow, microvascular rarefaction, inflammation, epigenetic aging

## Abstract

As aging societies in the western world face a growing prevalence of vascular cognitive impairment and Alzheimer’s disease (AD), understanding their underlying causes and associated risk factors becomes increasingly critical. A salient concern in the western dietary context is the high consumption of methionine-rich foods such as red meat. The present review delves into the impact of this methionine-heavy diet and the resultant hyperhomocysteinemia on accelerated cerebrovascular and brain aging, emphasizing their potential roles in cognitive impairment. Through a comprehensive exploration of existing evidence, a link between high methionine intake and hyperhomocysteinemia and oxidative stress, mitochondrial dysfunction, inflammation, and accelerated epigenetic aging is drawn. Moreover, the microvascular determinants of cognitive deterioration, including endothelial dysfunction, reduced cerebral blood flow, microvascular rarefaction, impaired neurovascular coupling, and blood–brain barrier (BBB) disruption, are explored. The mechanisms by which excessive methionine consumption and hyperhomocysteinemia might drive cerebromicrovascular and brain aging processes are elucidated. By presenting an intricate understanding of the relationships among methionine-rich diets, hyperhomocysteinemia, cerebrovascular and brain aging, and cognitive impairment, avenues for future research and potential therapeutic interventions are suggested.

## 1. Introduction

The Western world is undergoing a profound demographic shift, with an increasingly aging population [1,2,3]. This trend presents significant challenges, particularly regarding age-related cognitive decline [4,5]. Cognitive impairment and dementia have emerged as critical public health concerns, as they not only affect the well-being of individuals but also strain healthcare systems and social support networks [4,6,7,8,9]. With the prevalence of cognitive decline on the rise, gaining a comprehensive understanding of the underlying mechanisms contributing to this phenomenon and the lifestyle factors that influence it becomes of paramount importance.

Epidemiological studies have shed light on the prevalence and impact of cognitive decline and dementia across the aging populations of the developed world [10,11,12,13]. Alzheimer’s disease and other neurodegenerative disorders have received considerable attention due to their devastating effects on memory, cognition, and daily functioning [14,15]. However, emerging evidence suggests that vascular cognitive impairment and dementia (VCID) may be among the most significant contributors to age-related cognitive decline [16,17,18]. VCID is associated with cerebrovascular pathologies and functional impairment of the cerebral microcirculation, such as small vessel disease, dysregulation of cerebral blood flow (CBF), blood–brain barrier (BBB) disruption, cerebral microhemorrhages, and cerebral infarcts, which contribute to cognitive impairments independent of neurodegenerative changes [19,20,21,22,23,24,25,26,27,28,29,30,31,32,33,34].

Significant progress has been made in elucidating the cellular and molecular mechanisms underlying cerebrovascular and brain aging, as well as age-related pathologies affecting the central nervous system [28,29,35,36,37,38,39,40,41,42,43,44,45,46,47,48,49,50,51,52,53,54,55,56,57,58,59,60]. Accumulating evidence suggests that dietary factors play a crucial role in modulating the aging process [61,62,63,64,65] and the development of age-related diseases [66,67,68,69,70,71,72,73,74]. Importantly, dietary factors have also emerged as key determinants in shaping the trajectories of age-related cognitive decline and cerebrovascular and brain health [75]. Unhealthy dietary patterns, characterized by excessive consumption of high-fat, high-cholesterol, high-sugar, ultra-processed, and calorie-dense foods, are associated with detrimental effects on the brain, accelerating aging processes and increasing the risk of cognitive impairments [75].

Among dietary factors of interest, high methionine intake has been implicated as a potential contributor to unhealthy cerebrovascular and brain aging [76,77,78,79,80,81,82]. A methionine-rich diet refers to a dietary pattern that is high in methionine, an essential sulfur-containing amino acid. Methionine is an essential building block for protein synthesis and is involved in various biochemical processes. Beyond its role in protein synthesis, methionine also serves as a crucial methyl donor in cellular metabolism. Methionine is involved in the synthesis of S-adenosylmethionine (SAM), a universal methyl donor used in numerous methylation reactions. An excess intake of methionine, found abundantly in certain foods, can lead to increased oxidative stress, mitochondrial dysfunction, and inflammation when consumed in excess. Moreover, methionine metabolism can generate homocysteine, which has been associated with vascular dysfunction and cognitive impairments. Epidemiological studies have consistently highlighted the potential detrimental effects of high methionine consumption and hyperhomocysteinemia (a condition characterized by elevated levels of homocysteine, a metabolite of methionine, in the blood) on health outcomes, particularly in relation to age-related cardiovascular and cerebrovascular diseases and neurodegenerative disorders. Hyperhomocysteinemia is characterized by elevated levels of total homocysteine in the blood, exceeding the standard plasma homocysteine (Hcy) range of 5 to 15 μmol/L. For adults, the recommended daily allowance (RDA) for methionine is around 19 mg per kilogram of body weight, which translates to about 1.3 g for an average adult. A particular study highlighted a prevalence of 35.4% for this condition, with a notable gender disparity—45.4% in men compared to 28.5% in women [83]. Importantly, recent evidence links high methionine consumption and dysregulation of DNA methylation in relation to aging. The concept is emerging that dietary methionine intake plays an important role in modulation of epigenetic modification of DNA that regulates gene expression patterns, cellular function, and the aging process itself [84,85,86].

Preclinical studies utilizing laboratory animals have provided valuable insights into the effects of high-methionine diets and hyperhomocysteinemia on cardiovascular, cerebrovascular, and brain aging [77,87,88,89,90,91,92,93,94,95]. Animal models fed a methionine-rich diet have demonstrated accelerated aging-related changes, including increased vascular oxidative stress, mitochondrial dysfunction, inflammation, and impaired cognitive performance [77,87,88,89,90,91,92,93,94,95]. These studies have revealed the detrimental effects of high methionine intake on multiple facets of vascular and brain health and provide a foundation for further investigation into the underlying mechanisms. In contrast, animals on low methionine diets (“methionine restriction”) exhibit a youthful phenotype and increased health span, including attenuated oxidative stress, inflammation, and improved mitochondrial function [76,79,88,96,97,98,99,100,101,102,103,104,105,106,107,108,109,110,111,112]. These findings underscore the importance of understanding the impact of high and low methionine consumption on cerebrovascular health and its potential implications for cognitive impairment.

In light of these observations, this review aims to comprehensively examine the available evidence regarding the role of a methionine-rich diet in unhealthy cerebrovascular and brain aging, with a specific focus on its implications for cognitive impairment. By evaluating epidemiological data, preclinical studies, and the molecular mechanisms involved, we aim to provide a deeper understanding of the impact of high methionine consumption on cerebrovascular health, brain aging, and cognitive function. By examining the impact of a methionine-rich diet on cognitive decline and age-related pathologies, we aim to shed light on the importance of healthy dietary choices in supporting optimal brain health and mitigating the risk of cognitive impairment and dementia in aging populations. Understanding the impact of high methionine consumption on cerebrovascular and brain aging is essential for developing targeted dietary interventions to promote healthy aging and maintain cognitive function throughout the aging process.

## 2. Methionine-Rich Diet, Hyperhomocysteinemia, and Aging

### 2.1. Dietary Sources of Methionine

Methionine can be obtained through dietary sources. Certain foods are particularly high in methionine and can contribute to a methionine-rich diet. Foods that are known to be rich sources of methionine include red meat (beef, lamb, pork), fish (such as salmon, tuna), and dairy products (milk, cheese, yogurt), and some plant-based foods (e.g., legumes).

Red meat consumption in particular has been a topic of increasing interest in the context of aging and age-related diseases [113]. Access to fresh vegetables as compared to meat can vary based on factors like location, socioeconomic status, and individual circumstances. For instance, in developed countries like the U.S., meat constitutes a significant part of the diet, contributing to daily energy, protein, and fat intake due to its availability and preservation ease. On the other hand, fresh vegetables, crucial for a balanced diet, might not be as easily accessible in certain areas, which can be exacerbated by factors like neighborhood disparities and income levels. The accessibility of healthier food options, like fresh vegetables, is known to facilitate healthier diets and potentially lower the risk of overweight and obesity issues. Strategies to improve access to fresh vegetables, especially in comparison to meat and fast food outlets, are being explored to promote better nutrition and health outcomes across different communities [114]. Epidemiological studies have indicated that regular and high intake of red meat may be associated with accelerated aging processes and an increased risk of age-related diseases [115,116,117,118,119,120,121,122,123,124,125,126,127]. Of particular concern is the relationship between red meat consumption and cardiovascular and cerebrovascular diseases [120,121,123,127,128,129,130,131,132,133,134,135,136,137]. The potential adverse effects on health of red meat consumption have been attributed to an increased intake of cholesterol, saturated fat, and methionine [128,129,130,131,132,133,134,135,136,137]. Considering these factors, reducing red meat consumption and opting for healthier protein sources may have beneficial implications for the prevention of age-related cardiovascular and cerebrovascular diseases.

### 2.2. Methionine Metabolism

Methionine metabolism is a complex and highly regulated process that begins when methionine is taken up from food sources during digestion. Once ingested, methionine is absorbed into the bloodstream and transported to various tissues throughout the body. In the liver, the major site of methionine metabolism, methionine can undergo two main pathways. Through the transmethylation pathway, methionine is converted into SAM through a reaction catalyzed by the enzyme methionine adenosyltransferase. SAM is a universal methyl donor involved in various methylation reactions. It donates methyl groups to molecules such as DNA, histones, proteins, and neurotransmitters, playing a crucial role in epigenetic regulation, protein synthesis, and neurotransmitter metabolism. Excess SAM can be converted back to methionine through the methionine cycle.

Under normal conditions, a smaller portion of methionine is metabolized through the transsulfuration pathway. Here, methionine is converted into homocysteine, a process that requires the enzyme cystathionine beta-synthase (CBS). Subsequently, homocysteine is converted into cysteine, another important sulfur-containing amino acid. Normal methionine-homocysteine metabolism involves a tightly regulated series of biochemical reactions, including the remethylation pathway, that maintains the balance of these two sulfur-containing amino acids in the body. Folate and vitamin B12 are essential cofactors for the remethylation of homocysteine back to methionine. In this process, homocysteine is converted to methionine through the actions of methionine synthase, a vitamin B12-dependent enzyme. Folate donates a methyl group to homocysteine, which is transferred by methionine synthase, along with a methyl group from 5-methyltetrahydrofolate (the active form of folate), to produce methionine. Vitamin B6 is required for the transsulfuration pathway, where homocysteine is converted into cysteine through a series of reactions involving the enzyme cystathionine gamma-lyase (CGL). This well-coordinated system ensures that the levels of methionine and homocysteine are tightly regulated. Proper methionine-homocysteine metabolism is vital for various cellular processes, including protein synthesis, epigenetic regulation, neurotransmitter metabolism, and maintaining redox balance. Disruptions in this finely balanced metabolism, arising from deficiencies in the necessary vitamins, genetic mutations impacting the enzymes involved, or an excess intake of methionine, can lead to a condition known as hyperhomocysteinemia. This condition is associated with an elevated risk of various health issues, including cardiovascular and cerebrovascular diseases, neurodegenerative disorders, and other related health complications (as detailed below). 

### 2.3. Hyperhomocysteinemia

Hyperhomocysteinemia is a condition characterized by elevated levels of homocysteine, a sulfur-containing amino acid derived from methionine metabolism [77,138,139]. Homocysteine is normally metabolized through a process that requires specific vitamins, including folate, vitamin B12, and vitamin B6. The increased dietary intake of methionine, especially if combined with impairment of this metabolic pathway due to deficiencies in these vitamins or genetic variations affecting the enzymes involved in homocysteine metabolism, leads to hyperhomocysteinemia [138,139]. 

In addition to high methionine intake, hyperhomocysteinemia itself can affect various organ systems. Elevated homocysteine levels have been associated with an increased risk of cardiovascular diseases, including atherosclerosis, arterial thrombosis, and stroke [138,139,140,141,142,143,144,145,146]. Hyperhomocysteinemia has also been linked to cognitive impairment, neurodegenerative diseases, such as Alzheimer’s disease and Parkinson’s disease, and an increased risk of age-related macular degeneration [138,139]. Managing hyperhomocysteinemia involves addressing the underlying causes and optimizing vitamin intake. Supplementation with folate, vitamin B12, and vitamin B6 can help lower homocysteine levels in individuals with deficiencies or impaired metabolism. 

### 2.4. Effects of Methionine-Rich Diet and Hyperhomocysteinemia on Cellular Mechanisms of Aging

The mechanisms of aging represent a complex interplay of cellular processes that lead to the gradual decline of physiological functions and an increased susceptibility to age-related diseases. In recent years, research has shed light on the potential impact of dietary factors in general and methionine intake in particular on the aging process [76,79,88,96,97,98,99,100,101,102,103,104,105,106,107,108,109,110,111,112]. High methionine intake has been associated with detrimental effects on cellular homeostasis and overall health, promoting various cellular and molecular mechanisms of aging and the genesis of age-related diseases [76,79,88,96,97,98,99,100,101,102,103,104,105,106,107,108,109,110,111,112]. 

Conversely, evidence delineates a pleiotropic relationship between methionine intake and health outcomes. Methionine restriction—though potentially beneficial—can engender adverse effects including the onset of hair greying [147], atherosclerosis precipitated by improper homocysteine conversion [148], and, under severe restriction, the manifestation of fatty liver and anemia as demonstrated in rodent studies [149]. Furthermore, the utilization of methionine restriction as a therapeutic strategy in oncology is deemed impracticable due to the concomitant risks of muscle degradation and sarcopenia, thereby potentially hastening patient deterioration.

In this section, we explore some of the key mechanisms through which high methionine consumption can accelerate the aging process. From its role in increased oxidative stress, DNA damage, and induction of mitochondrial dysfunction to epigenetic modifications and altered protein homeostasis, we delve into the intricate pathways that link high methionine intake to the accelerated aging phenotype. Understanding these mechanisms is crucial for devising strategies to mitigate the adverse effects of a methionine-rich diet and potentially promote healthier aging outcomes.

#### 2.4.1. Oxidative Stress and Mitochondrial Dysfunction

Oxidative stress is a hallmark of aging and is implicated in the pathogenesis of various age-related diseases. Importantly, high methionine intake has been associated with increased cellular oxidative stress [93,150,151,152,153]. Conversely, dietary methionine restriction was shown to be associated with attenuated oxidative stress [88,107,108,109,110]. Methionine metabolism also produces toxic byproducts, such as homocysteine, leading to hyperhomocysteinemia, which also contributes to oxidative damage. The mechanisms by which high methionine intake and hyperhomocysteinemia exacerbate oxidative stress in aging likely include an imbalance between the increased production of reactive oxygen species (ROS) and the impaired ability of antioxidant systems to neutralize them [57,154,155,156,157]. 

There is strong evidence that high methionine intake and hyperhomocysteinemia increase mitochondrial ROS production, whereas dietary methionine restriction attenuates mitochondrial oxidative stress [88,110]. Mitochondria are crucial for energy production and play a vital role in cellular function. High methionine intake can disrupt mitochondrial function by increasing oxidative stress, impairing mitochondrial DNA integrity, and affecting mitochondrial dynamics. These disruptions can lead to mitochondrial dysfunction, compromised energy production, and bioenergetic impairment. Mitochondrial oxidative stress and mitochondrial dysfunction are hallmarks of aging and can contribute to various age-related diseases, including cardiovascular and cerebrovascular diseases and neurodegenerative disorders [44,45,48,51,54,55,60,158,159,160,161,162,163,164]. 

High methionine intake and/or hyperhomocysteinemia have also been shown to up-regulate major cellular oxidant systems, including NAD(P)H oxidases [90,91,92,93,94,95]. NAD(P)H oxidases, also known as NOX enzymes, are a family of enzymes contributing to age-related increases in ROS production in cells of the cardiovascular and immune systems as well as other organs [93,165,166]. In the context of high methionine intake and hyperhomocysteinemia, increased NAD(P)H oxidase activity can further exacerbate age-related oxidative stress [93,165]. Elevated homocysteine levels, in particular, have been associated with increased ROS production and oxidative damage in humans [167]. 

Both mitochondrial ROS generation and the activity and expression of cellular NAD(P)H oxidases are regulated by humoral factors [35,40,41,44,59,168,169,170] and proinflammatory cytokines, such as TNFα [93,171,172]. Previous studies have shown that in high-methionine-fed rodent models there is significant up-regulation of TNFα [93,173]. It is likely that increased inflammatory status associated with high methionine intake contributes to increased oxidative stress both in the cardiovascular system and the central nervous system [93,174,175,176,177,178]. High methionine intake and/or hyperhomocysteinemia were also reported to up-regulate xanthine oxidase [90] and cyclooxygenases [91,95]. Additionally, high methionine intake can also lead to the generation of ROS via the auto-oxidation of the sulfur atom in the SH-containing metabolites of methionine.

#### 2.4.2. Inflammation

Chronic low-grade inflammation, often referred to as inflammaging, is associated with accelerated aging and age-related cardiovascular, cerebrovascular, and brain diseases [36,37,54,179,180,181,182,183,184,185,186,187,188,189,190,191,192,193,194]. Methionine-rich diets have been shown to induce inflammation through multiple mechanisms, including the activation of pro-inflammatory signaling pathways and the generation of inflammatory mediators [78,99,108,153,195]. Chronic inflammation can contribute to accelerated cerebrovascular and brain aging by promoting increased oxidative stress and endothelial dysfunction, compromising mitochondrial energetics and increasing the susceptibility to microvascular damage and cerebrovascular senescence. A heightened state of inflammation also contributes to the pathogenesis of neurodegenerative disorders [181,191,196,197,198,199,200,201,202,203,204,205].

#### 2.4.3. Epigenetic Regulation of Aging Processes

Epigenetic modifications, including DNA methylation, histone modifications, and non-coding RNA expression, play a crucial role in regulating gene expression patterns, cellular function, genomic stability, cellular identity, and organismal aging [206,207,208,209,210,211,212,213,214,215,216,217]. DNA methylation, one of the major epigenetic mechanisms, involves the addition of a methyl group to DNA molecules, typically occurring at cytosine residues within CpG dinucleotides [208,211,212,218]. Methionine serves as a methyl donor in the synthesis of SAM, the principal supplier of methyl groups for DNA methylation reactions. Dysregulation of DNA methylation patterns is associated with accelerated aging [212,219] and is directly linked to the premature development of cognitive impairment and dementia [220,221,222]. High methionine intake can influence DNA methylation patterns, potentially contributing to the accelerated epigenetic aging [84,85,86,223,224,225]. Diet-induced changes in DNA methylation patterns can affect gene expression profiles, disrupt cellular functions, and increase susceptibility to age-related diseases. 

#### 2.4.4. DNA Damage and Repair Mechanisms

Increased oxidative stress leads to DNA damage accumulation in aged cells [226]. High methionine intake may contribute to increased DNA damage and impairments in DNA repair mechanisms [225]. Methionine metabolism generates reactive oxygen species (ROS) and reactive nitrogen species (RNS), which can lead to DNA damage [100,227]. Moreover, high methionine levels can disrupt the balance of one-carbon metabolism, affecting the availability of key molecules involved in DNA repair processes. Folate, a crucial B-vitamin, plays a pivotal role in DNA maintenance and repair, and its deficiency can contribute to DNA damage, especially when combined with high methionine intake [228]. Accumulated DNA damage and compromised repair mechanisms can accelerate the aging process and increase the risk of age-related diseases. DNA repair mechanisms play a pivotal role in maintaining genomic stability and preventing age-related phenotypic alteration (including induction of DNA-damage-mediated cellular senescence [191,215,229,230,231,232,233,234]) and functional decline. However, high methionine intake and hyperhomocysteinemia can interfere with these repair processes, contributing to DNA damage accumulation. Understanding the interplay between high methionine intake, hyperhomocysteinemia, and DNA damage/repair mechanisms is crucial for unraveling the mechanisms underpinning unhealthy cerebrovascular and brain aging. 

#### 2.4.5. Cellular Senescence

Cellular senescence is a cellular stress response, characterized by a state of irreversible growth arrest in which cells lose their ability to divide and proliferate, even in the presence of stimuli that would typically promote cell division. This state is often accompanied by distinct changes in cell morphology and function. Cellular senescence can be triggered by a variety of stressors, including DNA damage, telomere shortening, and exposure to oxidative stress. Senescent cells remain metabolically active but play a role in aging biology by secreting pro-inflammatory molecules and other factors that can influence tissue function and contribute to the pathogenesis of age-related diseases [46,57,157,235,236,237,238,239,240,241,242,243]. 

Importantly, cellular senescence has emerged as a pivotal player in the biology of cerebrovascular and brain aging, with profound implications for brain health [37,42,46,244]. Senescent cells accumulate over time in the brain and the cerebral microcirculation, and secrete a complex array of pro-inflammatory factors, collectively known as the senescence-associated secretory phenotype (SASP) [245]. This SASP can promote chronic inflammation, oxidative stress, and tissue dysfunction, all of which are central drivers of brain aging and the genesis of cognitive decline and neurodegenerative diseases. 

In the context of cerebrovascular and neurovascular aging, senescence of endothelial cells, pericytes, astrocytes, and paravascular microglia contributes to endothelial dysfunction, neurovascular dysfunction, blood brain barrier dysfunction, and genesis of cerebral microhemorrhages. These changes can lead to diminished cerebral blood flow, impairing nutrient and oxygen delivery to the aging brain, disruption of neuronal networks, and neuroinflammation. Furthermore, senescent cells within the cerebral circulation can disrupt neurovascular coupling, affecting the brain’s ability to respond to changing metabolic demands [42]. This, in turn, can contribute to cognitive decline and increased vulnerability to neurodegenerative diseases. The elimination of senescent cells [246] has recently been reported to improve lifespan and health span in rodents [247,248,249,250,251], consistent with the notion that senescent cells drive organismal aging. Importantly, clearance of senescent cells was shown to rejuvenate cerebromicrovascular function in mouse models of aging and accelerated brain aging [42,252,253]. 

Remarkably, a high-methionine diet has been implicated in the induction of cellular senescence [254,255,256]. Excessive methionine intake can lead to the generation of ROS, increased oxidative stress, and DNA damage, a known inducer of senescence. Moreover, methionine-derived homocysteine can promote endothelial dysfunction and inflammation, further exacerbating the senescent phenotype within the cerebral vasculature. Consequently, understanding the role of cellular senescence in vascular aging and its relationship with high methionine intake is of paramount importance. Efforts to detect senescent cells in animal models fed a high-methionine diet could involve profiling their transcriptomic signatures, utilizing techniques such as single-cell RNA sequencing [46], flow cytometry [253], or immunohistology. Furthermore, it is advisable to investigate the impact of senolytic treatments on these animal models, as this could yield valuable insights into the potential therapeutic effects of senolytics in the context of accelerated cerebrovascular and brain aging.

#### 2.4.6. Altered Protein Homeostasis and Proteostasis Network

Protein homeostasis, or proteostasis, is a critical cellular process responsible for maintaining the proper folding, function, and degradation of proteins, which is essential for maintaining cellular function and preventing protein aggregation, a hallmark of aging and neurodegenerative diseases [257,258,259,260,261,262,263,264]. High methionine intake may disrupt protein homeostasis by increasing oxidative stress and impairing the function of protein quality control mechanisms, including the ubiquitin-proteasome system and autophagy [265,266]. This disruption can lead to the accumulation of misfolded or damaged proteins, further contributing to cellular dysfunction and accelerated aging processes. Hyperhomocysteinemia can further exacerbate proteostatic disturbances [267]. Elevated homocysteine levels are associated with increased oxidative stress, which can promote protein oxidation and misfolding [268]. Homocysteine-induced oxidative modifications can lead to protein aggregates and impair the function of proteasomal and lysosomal degradation pathways, further compromising proteostasis.

## 3. Methionine-Rich Diet and Vascular Contributions to Cognitive Impairment

The brain’s elevated metabolic requirements are sustained by an intricate microcirculatory network, estimated to encompass approximately 600 km in length within the human body. This cerebral microcirculation serves as a vital conduit, ensuring the precise distribution of essential resources such as oxygen, glucose, and other nutrients to neural tissues. Additionally, it actively participates in the removal of metabolic waste products, maintains the ionic balance crucial for neuronal function, orchestrates the formation and integrity of the blood–brain barrier (BBB), and governs the transport of diverse substances across this barrier. Thus, the health of the microvasculature assumes a pivotal role in preserving normal cognitive and neuronal functions [20,21,45,47,49,50,51,269,270,271,272,273,274,275,276]. It is increasingly evident that dysfunction and damage within the cerebral microcirculation constitute significant contributors to age-related cognitive decline [17,29,273,274,275,277,278,279]. Clinical investigations have furnished evidence suggesting that the consumption of a methionine-rich diet, coupled with hyperhomocysteinemia, can incite dysregulation in cerebral blood flow, directly impacting cognition [280,281,282,283]. Experimental studies have extended these clinical observations, providing valuable mechanistic insights into the synergistic interplay between the aging process and diet-induced accelerated cellular aging, particularly concerning cerebromicrovascular function [173,178,284]. This section offers a comprehensive exploration of the distinct pathogenic roles played by endothelial dysfunction, neurovascular impairment, microvascular rarefaction, and blood–brain barrier disruption in the genesis of vascular cognitive impairment (VCI). Furthermore, we scrutinize the potential involvement of a methionine-rich diet and hyperhomocysteinemia in exacerbating these pathogenic processes, shedding light on the intricate interconnections between diet, vascular health, and cognitive outcomes.

### 3.1. Atherosclerosis and Stroke

Both hyperhomocysteinemia and increased methionine intake are associated with atherosclerotic vascular diseases, including stroke [140,141,142,143,144,145,146,285,286,287]. Previous studies also confirmed significant genetic associations between premature ischaemic stroke and haplotypes in various genes involved in methionine metabolism [285,288,289]. Preclinical studies distinguished between the effects of excess dietary methionine from those of genetic forms of hyperhomocysteinemia and provides evidence that consumption of a methionine-rich diet per se promotes atherogenesis [87,290,291,292,293,294,295].

### 3.2. Endothelial Dysfunction and Dysregulation of Cerebral Blood Flow

Endothelial cells play a crucial role in maintaining vascular homeostasis. Endothelium-dependent, NO-mediated dilation of cerebral microvessels plays a crucial role both in maintaining cerebral blood flow [279,296] and mediating neurovascular coupling responses [40,47,48,55,154,252,297,298,299,300,301]. Neurovascular coupling refers to the tight regulation of cerebral blood flow in response to neuronal activity. This mechanism ensures that an adequate supply of oxygen and nutrients is delivered to active brain regions.

With aging, endothelial dysfunction occurs, characterized by impaired endothelium-dependent vasodilation, increased oxidative stress, and inflammation [24,42,44,47,48,51,52,55,57,59,162,168,171,183,269,270,271,301,302,303,304,305,306]. This dysfunction contributes to cognitive decline by compromising cerebral blood flow regulation, impairing nutrient and oxygen delivery to the brain, and disrupting the blood–brain barrier (BBB) [20,21,27,30,43,47,48,55,269,272,273,274,279,307,308,309]. Clinical studies have shown associations between endothelial dysfunction and cognitive impairment in aging populations [24,302]. Age-related impairments in neurovascular coupling, due to endothelial dysfunction, have been observed and are associated with cognitive dysfunction [21,35,42,47,48,51,55,270]. 

Earlier research has indicated that consumption of a methionine-rich diet and an increase in plasma homocysteine levels can impair endothelium-mediated vasodilation in humans [150,310,311,312,313,314,315,316], mimicking the aging phenotype. Additionally, preclinical studies have demonstrated that high-methionine diets or hyperhomocysteinemia can exacerbate endothelial dysfunction [90,91,92,93,94,95,317]. This occurs through a reduction in NO bioavailability due to increased vascular oxidative stress [90,91,92,93,94]. Preliminary preclinical evidence suggests that high-methionine diets and hyperhomocysteinemia may also disrupt the synthesis of vasoactive arachidonic acid metabolites within the vasculature [91,95,306]. Endothelial dysfunction induced by high-methionine diets or hyperhomocysteinemia can further disrupt neurovascular coupling by impairing the dilation of cerebral arterioles in response to neuronal activation [281,318]. Clinical and preclinical studies have reported associations between high methionine intake or hyperhomocysteinemia, dysregulation of cerebral blood flow, and cognitive dysfunction [173,178,280,282,283,284,319].

### 3.3. Microvascular Rarefaction

Microvascular rarefaction refers to a reduction in the density and branching of small blood vessels, impairing microcirculation and nutrient exchange in the brain. Aging is associated with microvascular rarefaction, which can compromise cerebral perfusion and contribute to cognitive decline [26,43,58,279,301,320,321]. Crucially, there exists a direct correlation between the degree of age-related capillary rarefaction observed in the hippocampus and the severity of cognitive decline. This correlation serves as supplementary evidence affirming the intimate link between the disruption of cerebral blood flow and impaired neuronal function. High-methionine diets or hyperhomocysteinemia have been shown to accelerate microvascular rarefaction, promoting the loss of capillaries and impairing the integrity of the neurovascular unit [77]. The mechanisms that potentially contribute to cerebromicrovascular rarefaction in the context of aging may encompass several factors, including diminished endothelial senescence, endothelial NO availability [322,323,324,325], pericyte depletion [301], heightened endothelial cell apoptosis [326,327], reduced levels of pro-angiogenic factors like VEGF [328] and IGF-1 [44,58,329,330,331], and compromised endothelial angiogenic processes [58,159,301,332,333,334,335]. Further studies are warranted to determine how consumption of a diet rich in methionine impacts these synergistic mechanisms in the cerebral microcirculation. 

### 3.4. Blood–Brain Barrier Disruption and Neuroinflammation

The blood–brain barrier (BBB) regulates the exchange of substances between the blood and the brain, maintaining brain homeostasis [29,273,274]. Aging is associated with BBB dysfunction, characterized by increased permeability and compromised barrier integrity [43,204,309,336,337,338,339]. BBB disruption can lead to neuroinflammation, neuronal damage, and cognitive impairment [29,30,273,274,309]. Preclinical evidence suggests that high-methionine diets or hyperhomocysteinemia may contribute to BBB disruption, allowing the entry of harmful substances into the brain and exacerbating neuroinflammation and cognitive decline [78,284,318,340,341,342,343]. Understanding the potential mechanisms underlying BBB disruption induced by high methionine intake and hyperhomocysteinemia and its role in accelerated brain aging is crucial for identifying potential therapeutic interventions that target cerebrovascular health and mitigate the detrimental effects of high methionine consumption on cognitive function. The mechanisms contributing to increased BBB permeability are likely to be multifaceted. Firstly, there may be alterations in the expression levels of tight junction and adherens junction proteins potentially compromising BBB integrity [344,345]. Moreover, high methionine intake and hyperhomocysteinemia are expected to induce post-translational modifications of tight junction proteins. These modifications could impact the stability and appropriate cellular localization of these proteins. Pericytes, crucial structural components of the BBB, play a pivotal role in maintaining its integrity. In this context, it’s noteworthy that animal models of hyperhomocysteinemia exhibit pericyte damage [344,345]. There is also increasing evidence that cellular senescence promotes BBB disruption [244,252,346,347]. Additionally, the cells that constitute the BBB exhibit a high metabolic rate, consistent with their elevated energy demands to support the activities of ATP-dependent transporters. Both high methionine intake and hyperhomocysteinemia were shown to compromise mitochondrial function in the vasculature [100,174,175]. 

A significant outcome of BBB disruption is the leakage of plasma components, including IgG, thrombin, and fibrinogen, into the brain parenchyma [348]. This heightened infiltration of plasma proteins across the BBB serves as a catalyst for neuroinflammation, primarily driven by the activation of resident immune cells, particularly microglia [348]. Notably, the increased presence of activated microglia in the hippocampi is closely linked to the aggravated impairment of long-term potentiation (LTP) in excitatory synaptic transmission. LTP is a fundamental cellular mechanism associated with learning and memory [349].

## 4. Methionine-Rich Diet and Synaptic Function/Neuronal Health

### 4.1. Effects of High Methionine Intake and Hyperhomocysteinemia on Synaptic Plasticity and Neurotransmitter Systems

Synaptic plasticity, the ability of synapses to strengthen or weaken over time in response to changes in their activity, is fundamental to learning and memory processes. High methionine intake and hyperhomocysteinemia have been shown to impact synaptic plasticity by disrupting key molecular signaling pathways involved in synaptic function [223,350,351,352,353,354]. Preclinical studies have demonstrated that a methionine-rich diet can impair long-term potentiation (LTP), a cellular mechanism underlying synaptic plasticity, leading to deficits in learning and memory [351,354,355]. Elevated homocysteine levels, stemming from high methionine intake, have been linked to reduced levels of brain-derived neurotrophic factor (BDNF) [356], a molecule essential for synaptic plasticity. Reduced BDNF impairs LTP. Furthermore, homocysteine can induce oxidative stress, further inhibiting LTP and leading to synaptic dysfunction. High methionine intake and hyperhomocysteinemia can also influence various neurotransmitter systems [357]. Elevated homocysteine levels have been associated with disruptions in the dopaminergic, serotonergic, and cholinergic neurotransmitter systems. These disturbances can lead to mood disorders, cognitive deficits, and other neurological issues.

### 4.2. Neuronal Damage and Neurodegenerative Processes Induced by High Methionine Intake and Hyperhomocysteinemia 

Neurodegenerative diseases, characterized by the progressive loss of structure and function of neurons, represent a burgeoning global health concern [6,358]. Conditions such as Alzheimer’s disease and Parkinson’s disease have seen a notable increase in prevalence, primarily due to the aging global population. Alzheimer’s disease, in particular, stands out as the most common form of dementia, affecting millions worldwide, with its incidence expected to triple by 2050. These diseases not only inflict immense emotional and physical burdens on patients and caregivers but also pose significant socioeconomic challenges. The rising prevalence underscores the urgency of understanding their pathophysiology, developing effective preventive strategies, including dietary interventions. 

Preclinical studies have shown that a methionine-rich diet and hyperhomocysteinemia can induce neurodegenerative disease-like changes, including protein aggregation and neuroinflammation. Clinical evidence has also suggested associations between high methionine intake or hyperhomocysteinemia, imaging biomarkers of accelerated aging of the central nervous system [359,360], and an increased risk of neurodegenerative diseases, such as Alzheimer’s disease and Parkinson’s disease. A sustained high-methionine diet and hyperhomocysteinemia have been implicated in direct and indirect neuronal damage [77,81,176,352,361,362]. Oxidative stress is one mechanism by which high methionine intake and hyperhomocysteinemia induces cellular damage. Elevated levels of homocysteine increase the production of reactive oxygen species, leading to oxidative stress and subsequent damage to neuronal DNA, proteins, and lipids. Furthermore, a high-methionine diet and hyperhomocysteinemia has been linked to mitochondrial dysfunction in neurons. Mitochondrial dysfunctions can trigger a cascade of events, including the release of pro-apoptotic proteins, leading to neuronal cell death. Moreover, homocysteine can activate microglia, leading to neuroinflammation, another mechanism contributing to neurodegeneration [342,363]. Besides, there are indications that high methionine levels might influence the formation and aggregation of beta-amyloid plaques, a hallmark of Alzheimer’s disease, further suggesting a link between methionine-rich diets and neurodegenerative processes [363,364,365,366,367,368,369,370,371,372,373,374,375]. Elevated homocysteine levels have also been associated with increased tau hyperphosphorylation, a hallmark of Alzheimer’s disease [176,370]. Additionally, studies have shown a correlation between hyperhomocysteinemia and enhanced production of amyloid-beta, the primary component of amyloid plaques in Alzheimer’s disease [89,373].

### 4.3. Methionine-Rich Diet, Hyperhomocysteinemia, and Altered Functional Connectivity

Functional connectivity refers to the temporal correlation of neural activity between different brain regions, which reflects the coordination and integration of neural networks. Disruptions in functional connectivity with aging [376] have been implicated in various cognitive impairments and neurological disorders [377,378,379,380,381,382,383,384,385,386,387,388]. Brain connectivity, essential for efficient cognitive function, can be influenced by dietary factors [389,390,391]. In particular, hyperhomocysteinemia has been shown to influence functional connectivity patterns in the brain [82,392]. These studies suggest that hyperhomocysteinemia can alter the default mode network, a key brain network active during rest and implicated in memory processes. These alterations in functional connectivity may contribute to cognitive impairments observed in individuals consuming a methionine-rich diet. The exact mechanisms underlying the effects of high methionine intake on functional connectivity are still being explored. However, it is believed that oxidative stress, inflammation, and mitochondrial dysfunction induced by high methionine consumption can cause white matter damage [368,393] and disrupt the normal communication and coordination between brain regions, leading to altered functional connectivity. Understanding the impact of a methionine-rich diet on functional connectivity provides further insights into the potential mechanisms through which high methionine intake may contribute to cognitive impairments. 

## 5. Conclusions and Perspectives

In summary, the relationship between elevated methionine intake, hyperhomocysteinemia, and the aging processes affecting cerebrovascular and brain health stands as a pivotal area of investigation in understanding age-related cognitive decline. This review has shed light on the multifaceted pathways through which these dietary and metabolic factors accelerate aging mechanisms within the cerebrovascular system and brain, ultimately contributing to cognitive impairment. From the initiation of oxidative stress and mitochondrial dysfunction to the induction of chronic inflammation and accelerated epigenetic aging, high methionine consumption and hyperhomocysteinemia set forth a cascade of detrimental effects.

The body of evidence available underscores the pivotal role played by microvascular health in the context of age-related cognitive impairment, offering illuminating insights into how factors such as endothelial dysfunction, reduced cerebral blood flow, microvascular rarefaction, compromised neurovascular coupling, and blood–brain barrier disruption collectively contribute to the complex landscape of cognitive decline. Importantly, it should be noted that these factors are modifiable through dietary interventions and lifestyle adjustments, providing promising avenues for the preservation of cognitive function and the promotion of healthy brain aging. The profound impact on synaptic function and neuronal health further underscores the extensive consequences of these dietary factors.

Looking forward, further research should delve deeper into the intricate molecular mechanisms underpinning the relationship between methionine-rich diets, hyperhomocysteinemia, and cerebrovascular and brain aging. It is imperative to investigate the dose-response relationship between methionine intake and cerebrovascular/brain aging to establish optimal intake thresholds. Additionally, efforts should focus on elucidating the molecular mechanisms through which high methionine intake affects microvascular dysfunction, synaptic function, and neuronal health. Exploring the interactions between methionine metabolism and other dietary factors in modulating cerebrovascular health and brain aging will be crucial in gaining a comprehensive understanding of these processes. Investigations into potential interventions to mitigate the detrimental effects of high methionine intake on cerebrovascular health and brain aging are warranted. Strategies targeting these mechanisms, such as dietary modifications, supplementation, or pharmacological interventions, should be further explored in preclinical and clinical settings.

To optimize cerebrovascular and brain health and counteract the adverse effects of a methionine-rich diet, several dietary recommendations and potential therapeutic strategies can be considered [394,395,396,397,398,399]. These include optimizing methionine intake by adopting a balanced diet rich in nutrient-dense whole foods, favoring plant-based diets [400] like the mediterranean diet [401] or vegetarian/vegan diets. Such diets naturally limit methionine intake while providing essential amino acids. Increasing folate intake, found in foods like leafy greens and legumes, can aid in homocysteine metabolism. An antioxidant-rich diet comprising fruits, vegetables, nuts, and seeds can help combat oxidative stress linked to high methionine intake. Regular physical exercise has demonstrated benefits in improving cerebrovascular function, reducing oxidative stress, and enhancing cognitive performance [402]. Moreover, pharmacological interventions, such as antioxidants, anti-inflammatory agents, or compounds targeting mitochondrial function, may hold promise for mitigating the detrimental effects of high methionine intake on cerebrovascular and brain aging. Nevertheless, further research is essential to validate the effectiveness and safety of these interventions in clinical contexts.

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
