# Peer review of "The Role of Methionine-Rich Diet in Unhealthy Cerebrovascular and Brain Aging: Mechanisms and Implications for Cognitive Impairment"

_nutrients, 2023, doi:10.3390/nu15214662_

Round 1

Reviewer 1 Report

Comments and Suggestions for Authors

Ungvari et al present a comprehensive and timely review of methionine and its potential impact on vascular aging in the brain. The work is quite complete and very well written. Overall, I have no overwhelming concerns. I would respectfully ask the author’s to consider a figure or schematic, perhaps representing the overall metabolism of the methionine and how excess methionine consumption drives dysfunction. It would just provide clarity regarding the overall pathways that could be interrupted, but is certainly not required for acceptance.

Minor Thoughts:

1.       There’s an odd dash that comes up in the abstract, in the words “disrup-tion” and “intri-cate” and “cerebrovas-cular” and “in-terventions”

2.       I believe that hyperhomocysteinemia has a specific definition regarding plasma levels in humans, it might be worthwhile mentioning this, as well as including a brief sentence on overall numbers of patients afflicted.

3.       Along those lines, it might be worthwhile to consider adding more specific details regarding diet overall, like how many people have access to excessive meat (especially given how easy it is to preserve) but not to fresh vegetables. Just a thought.

Author Response

We extend our sincere gratitude to the reviewers and the Editor for their thoughtful and constructive assessment of our work. We are truly appreciative of your positive feedback, and it is gratifying to know that you found our review to be comprehensive and well-crafted. Your valuable suggestion to incorporate a figure or schematic illustrating methionine metabolism and its potential implications for vascular aging in the brain has not gone unnoticed. Your feedback has significantly enriched the quality of our manuscript, and we eagerly anticipate your evaluation of the revisions we have made to address your recommendations in the updated version. Thank you once again for your invaluable insights.

Reviewer 1:

Ungvari et al present a comprehensive and timely review of methionine and its potential impact on vascular aging in the brain. The work is quite complete and very well written. Overall, I have no overwhelming concerns. I would respectfully ask the author’s to consider a figure or schematic, perhaps representing the overall metabolism of the methionine and how excess methionine consumption drives dysfunction. It would just provide clarity regarding the overall pathways that could be interrupted, but is certainly not required for acceptance.

Minor Thoughts:

There’s an odd dash that comes up in the abstract, in the words “disrup-tion” and “intri-cate” and “cerebrovas-cular” and “in-terventions”

Our Answer: This has been corrected, thank you.

I believe that hyperhomocysteinemia has a specific definition regarding plasma levels in humans, it might be worthwhile mentioning this, as well as including a brief sentence on overall numbers of patients afflicted.

Our Answer: We thank the reviewer for the valuable feedback. Indeed, hyperhomocysteinemia does have specific plasma level definitions in humans, typically referring to elevated homocysteine levels in the blood. We appreciate your suggestion and have included this information in our reviewed manuscript to provide more clarity for readers. Additionally, incorporating a brief sentence on the prevalence or overall numbers of patients affected by hyperhomocysteinemia is an excellent idea, as it offers a broader perspective on the relevance and importance of this condition.

Along those lines, it might be worthwhile to consider adding more specific details regarding diet overall, like how many people have access to excessive meat (especially given how easy it is to preserve) but not to fresh vegetables. Just a thought.

Our Answer: It's a valuable point to address the accessibility and dietary patterns related to excessive meat consumption versus fresh vegetables. We agree that providing more specific details about these aspects further enriches our review by offering a deeper understanding of the factors contributing to methionine intake and its potential impact on health. We have incorporated relevant information into the manuscript to provide a more comprehensive perspective on the dietary factors involved in methionine consumption.

Reviewer 2 Report

Comments and Suggestions for Authors

Review nutrients-2675068

In this manuscript the authors review the role of methionine-rich diet in unhealthy cerebrovascular and brain aging.

The reviewers state in the introduction that: “An excess intake of methionine, found abundantly in certain foods, can lead to increased oxidative stress, mitochondrial dysfunction, and inflammation when consumed in excess”. What is ‘excess’? Are there examples in literature that show that excess intake is something common in humans? And is there human proof that increased methionine intake is directly causal to disease?

What is ‘high methionine consumption’? The authors state that a broad range of foods are considered methionine-rich (meat, fish, dairy and vegetables), but focus on the papers that assessed red meat consumption. In my opinion, many of the ‘red meat-papers’ were to much focussed on blaming red meat consumption that they forgot to look at deficiencies (eg. vitamins and minerals) that are caused by one-sided diets. In that view, are we looking at methionine overload or more at “hidden” deficiencies?

What happens with excess methionine in the human body? Is it actively excreted? Do we find increased traces in urine or faeces?

Is hyperhomocysteinemia primary a result of increased cysteine intake or methionine intake? And what is the additive effect of deficiencies in specific vitamins (folate, vitamin B12 and B6)? Is hyperhomocysteinemia primary a result of excess intake or ‘hidden deficiency’?

Apparently the effect of methionine intake on health outcome follows a U-shape. The review focusses on increased methionine intake, but the review would benefit from a paragraph that puts the balance in perspective. I guess that a part of the population would actually benefit from methionine fortification of their diet. An additional figure would greatly help with the readability.

Also a figure that shows the causality of excess methionine in the presentation/molecular pathways of ‘Oxidative stress and mitochondrial dysfunction’, ‘Inflammation’, ‘Epigenetic regulation’, etc. would benefit the reader. Now it is mostly associations and not causal. It also does not help that it is not clear what ‘excess’ is and what the ‘normal values’ are.

In the examples brought up by the authors, do the methionine-rich diets represent increased human consumption as observed in population-cohorts or excessive intake (as is often the case in animal studies, making them unrepresentable for the human situation?

Author Response

Reviewer 2:

In this manuscript the authors review the role of methionine-rich diet in unhealthy cerebrovascular and brain aging.

The reviewers state in the introduction that: “An excess intake of methionine, found abundantly in certain foods, can lead to increased oxidative stress, mitochondrial dysfunction, and inflammation when consumed in excess”. What is ‘excess’? Are there examples in literature that show that excess intake is something common in humans? And is there human proof that increased methionine intake is directly causal to disease?

Our Answer: The term "excess" in the context of methionine intake can vary depending on individual factors, including age, sex, and overall dietary patterns. Generally, excess methionine intake typically refers to consumption significantly above the recommended daily intake levels. For adults, the recommended daily allowance (RDA) for methionine is around 19 mg per kilogram of body weight, which translates to about 1.3 grams for an average adult. Exceeding this recommended amount on a regular basis may be considered "excess."

What is ‘high methionine consumption’? The authors state that a broad range of foods are considered methionine-rich (meat, fish, dairy and vegetables), but focus on the papers that assessed red meat consumption. In my opinion, many of the ‘red meat-papers’ were to much focussed on blaming red meat consumption that they forgot to look at deficiencies (eg. vitamins and minerals) that are caused by one-sided diets. In that view, are we looking at methionine overload or more at “hidden” deficiencies?

Our Answer: The reviewer’s perspective raises an important consideration when evaluating the impact of methionine consumption, especially in the context of studies that primarily focus on red meat consumption. While it's essential to examine the potential effects of high methionine intake, it's equally crucial to consider the broader dietary context and the potential for "hidden" deficiencies in a one-sided diet. Deficiencies in vitamins and minerals, such as iron, vitamin B12, and zinc, can certainly result from one-sided diets, and these deficiencies can have significant health implications. Thus, it's essential to take a holistic approach when studying dietary components like methionine or red meat and consider not only potential "overload" but also the possibility of "hidden" deficiencies that can result from imbalanced diets.

What happens with excess methionine in the human body? Is it actively excreted? Do we find increased traces in urine or faeces?

Our Answer: Excess methionine in the human body is primarily metabolized and excreted through several processes, with the majority of it being converted to other compounds or used in various biochemical reactions. 1. conversion of methionine to homocysteine, which can then be further metabolized into cysteine or processed through the transsulfuration pathway. A portion of excess methionine and its metabolites may be excreted in urine as waste products. Homocysteine, for example, can be excreted in the urine when present in excessive amounts.

Is hyperhomocysteinemia primary a result of increased cysteine intake or methionine intake? And what is the additive effect of deficiencies in specific vitamins (folate, vitamin B12 and B6)? Is hyperhomocysteinemia primary a result of excess intake or ‘hidden deficiency’?

Our Answer: Hyperhomocysteinemia primarily results from disturbances in methionine metabolism, specifically the conversion of methionine to cysteine. It is not primarily a result of increased cysteine intake but rather issues related to methionine metabolism. Methionine intake and Vitamin deficiencies can contribute to hyperhomocysteinemia: a high-methionine diet can contribute to increased homocysteine levels. Deficiencies in specific B vitamins, particularly folate (vitamin B9), vitamin B12 (cobalamin), and vitamin B6 (pyridoxine), play a significant role in methionine metabolism as these vitamins are essential cofactors for enzymes involved in homocysteine metabolism. When there is a deficiency in these vitamins, homocysteine is not efficiently converted to cysteine or other metabolites, leading to its accumulation in the blood.

Apparently the effect of methionine intake on health outcome follows a U-shape. The review focusses on increased methionine intake, but the review would benefit from a paragraph that puts the balance in perspective. I guess that a part of the population would actually benefit from methionine fortification of their diet. An additional figure would greatly help with the readability.

Our Answer: The reviewer’s suggestion to include a paragraph that provides a balanced perspective on the effects of methionine intake and its potential benefits was very interesting and valuable, this addition enhanced the comprehensibility of the review. Methionine is involved in the production of S-Adenosyl Methionine (SAM-e), which is used to improve mood and well-being, treat depression and arthritis, and detoxify the liver. These benefits might be relevant for elderly individuals dealing with mood disorders, arthritis, or liver issues, although the data on SAM-e's efficacy are conflicting

Also a figure that shows the causality of excess methionine in the presentation/molecular pathways of ‘Oxidative stress and mitochondrial dysfunction’, ‘Inflammation’, ‘Epigenetic regulation’, etc. would benefit the reader. Now it is mostly associations and not causal. It also does not help that it is not clear what ‘excess’ is and what the ‘normal values’ are.

Our Answer: Thank you for the suggestion, we have included an illustration of the pathways between excess methionine and various health outcomes, such as oxidative stress, mitochondrial dysfunction, inflammation, and epigenetic regulation.

In the examples brought up by the authors, do the methionine-rich diets represent increased human consumption as observed in population-cohorts or excessive intake (as is often the case in animal studies, making them unrepresentable for the human situation?

Our Answer: The interpretation of methionine-rich diets in research can indeed vary depending on the context of the study. In most cases, methionine-rich diets are designed to simulate increased human consumption as observed in population cohorts.

Round 2

Reviewer 2 Report

Comments and Suggestions for Authors

I am very pleased with the answers given by the authors and have no further comments.